# Transport Properties for Pharmaceutical Controlled-Release Systems: A Brief Review of the Importance of Their Study in Biological Systems

**DOI:** 10.3390/biom8040178

**Published:** 2018-12-17

**Authors:** Ana C. F. Ribeiro, Miguel A. Esteso

**Affiliations:** 1Department of Chemistry, Coimbra Chemistry Centre, University of Coimbra, 3004-535 Coimbra, Portugal; 2U.D. Química Física, Universidad de Alcalá, Alcalá de Henares, 28871 Madrid, Spain

**Keywords:** solutions, transport properties drug–carriers, multicomponent systems

## Abstract

The goal of this work was to comprehensive study the transport properties of controlled-release systems for the safe and reliable delivery of drugs. Special emphasis has been placed on the measurement of the diffusion of drugs, alone or in combination with carrier molecules for enhanced solubility and facilitated transport. These studies have provided detailed comprehensive information—both kinetic and thermodynamic—for the design and operation of systems for the controlled release and delivery of drugs. Cyclodextrins are among the most important carriers used in these systems. The basis for their popularity is the ability of these materials to solubilize poorly soluble drugs, generally resulting in striking increases in their water solubilities. The techniques used in these investigations include pulse voltammetry, nuclear magnetic resonance (NMR) and Raman spectroscopy, ultrasonic relaxation, and dissolution kinetics. Transport in these systems is a mutual diffusion process involving coupled fluxes of drugs and carrier molecules driven by concentration gradients. Owing to a strong association in these multicomponent systems, it is not uncommon for a diffusing solute to drive substantial coupled fluxes of other solutes, mixed electrolytes, or polymers. Thus, diffusion data, including cross-diffusion coefficients for coupled transport, are essential in order to understand the rates of many processes involving mass transport driven by chemical concentration gradients, as crystal growth and dissolution, solubilization, membrane transport, and diffusion-limited chemical reactions are all relevant to the design of controlled-release systems. While numerous studies have been carried out on these systems, few have considered the transport behavior for controlled-release systems. To remedy this situation, we decided to measure mutual diffusion coefficients for coupled diffusion in a variety of drug–carrier solutions. In summary, the main objective of the present work was to understand the physical chemistry of carrier-mediated transport phenomena in systems of controlled drug release.

## 1. Introduction

### 1.1. Literature Review

In the past two decades, our research group and others [1,2,3,4,5,6,7,8,9] have studied the transport properties of systems involving drugs, motivated by their practical contribution to a better understanding of the mechanism of drug release. The work developed by these authors [1,2,3,4,5,6,7,8,9] has been focused on multicomponent diffusion including coupled transport, by using different methods (i.e., Taylor’s method with acceptable uncertainty of 1%–2%, and the interferometric methods with uncertainty <0.5%). In fact, both methods have also been used to determine multicomponent diffusion coefficients for ternary systems relevant to diffusion-based controlled release. However, the present work will focus on Taylor’s method. Some results have been obtained for systems involving drugs used in the treatment of both Parkinson’s disease and tuberculosis, in different media at therapeutic dosages [2,10,11]. These measurements and others (e.g., some quaternary [12], ternary [13,14,15,16,17], and binary systems [18,19,20,21,22]) were obtained in our laboratory by using the Taylor dispersion method. From these data, it was also possible to estimate some other parameters (e.g., diffusion coefficients at infinitesimal concentrations, dissociation degrees, activation energies, activity coefficients, and effective hydrodynamic radii). In addition, using ternary diffusion coefficients, we can conclude if the presence of a certain solute affects the diffusion of another one. On the other hand, using the open-ended capillary cell, the Ribeiro group has also focused on the study of the diffusion behaviour of pseudo-binary systems such as sodium dodecyl sulfate (SDS)/sucrose/water [23], SDS/cyclodextrin/water [24]. 

Leaist and his group have extensively studied the diffusion of different systems using a variety of techniques, including the Taylor dispersion techniques, dynamic light scattering, thermogravitation cells, and computer simulations [25,26,27,28,29,30,31,32]. The results have provided new information about both molecular motions and interactions, in order to understand the rates of chemical and physical processes of practical significance, such as diffusion-limited reactions, carrier-mediated transport, solubilization, gas absorption, crystal growth, chemical waves and oscillations, and diffusion driven by temperature gradients (i.e., the Soret effect). Some of the systems studied were protein solutions with added salt and buffer electrolytes and mixed micelle, micelle + solubilizer, and micelle + polymer solutions. Diffusion in these systems and many others is characterized by strongly coupled diffusion. This means that a gradient in the concentration of a substance can drive substantial coupled diffusion flows of other substances. Leaist’s group has shown, for example, that a mole of diffusing protein can transport hundreds of moles of salts or buffer electrolytes. In polymer + surfactant solutions, such as aqueous poly(ethylene glycol) + sodium dodecylsulfate, thousands of moles of surfactant can be transported per mole of diffusing polymer. More surprisingly, they demonstrated “incongruent” diffusion for solutions of mixed surfactant micelles, that is, a surfactant can be driven from lower to higher concentration (uphill) by its own concentration gradient. A key part of their research has been the development of predictive models and the interpretation of multicomponent diffusion properties. They have devoted considerable effort toward understanding the mechanism of coupled transport in terms of molecular association, nonideal solution behaviour, and ion migration driven by diffusion potential gradients.

Concerning self-diffusion, *D** [25], many techniques are used to study it in aqueous solutions. Methods such as NMR, polarography, and capillary-tube techniques with radioactive isotopes measure self-diffusion coefficients (intradiffusion coefficients). However, relationships derived between intradiffusion coefficients, *D**, and mutual diffusion coefficients, *D*, have had limited success, thus necessitating experimental mutual diffusion coefficients. 

Studies involving drugs show that their therapeutic efficacy is still a largely unknown territory because there are too many variables that are difficult to control and have not yet been studied. Thus, by combining the expertise relevant to the pharmaceutical applications of different research groups, it has been possible to reach a better comprehensive understanding of the thermodynamic and coupled transport properties of systems involving drugs and carriers for controlled release systems. In this manner, the physical chemistry groups at the University of Coimbra (Portugal) have collaborated with the group at St Francis Xavier University (Canada) to make high-precision binary and multicomponent diffusion measurements for different systems, including solutions of surfactant micelles. This work has provided fundamental new information about the properties of micelle carrier systems, which can be useful in different applications in the biomedical, pharmaceutical, and environmental fields. In addition, important efforts have been dedicated to the analysis of the results and the development of theoretical models to explain coupled diffusion behaviour and the molecular structures in associating systems. Moreover, during the past few years, additional collaboration between the physical chemistry groups of the universities of Coimbra (Portugal) and Alcalá (Spain) has allowed us to make significant research contributions concerning the thermodynamic and transport properties of systems involving drugs and carriers [2,10]. Thus, through the combined and complimentary expertise of this international macro-team in transport and thermodynamic properties, spectroscopy [33,34,35], chemical and mechanical characterization, and theoretical calculations, it has been possible to obtain a better understanding of complex systems involving carriers such cyclodextrins and surfactants for controlled drug release.

### 1.2. Methods 

Physiochemical and toxicological properties limit the benefits of drugs administered in their traditional ways. Developing drug release systems has allowed for the manipulation of some of these properties, improving therapeutic effects and favouring their clinical use. The aim of this introductory chapter is to present the ways used to evaluate both transport and thermodynamic properties for systems involving drugs—mainly the drugs used in the treatment of Parkinson’s and tuberculosis diseases—with cyclodextrins at therapeutic dosages, and at different temperatures and media, in order to represent physiological conditions. In the next chapters, the corresponding numerical data obtained will be shown.

The Taylor dispersion technique will be used for mutual diffusion measurements in solutions of carriers (emphasizing cyclodextrins and surfactants such as pluronics and sodium dodecylsulfate) and different drugs (e.g., l-dopa and ethambutol dihydrochloride), alone or in combination with others, in different molar ratio, taking advantage of the extensive research experience of our team in the measurement and interpretation of diffusion processes. In this way, we intend to contribute not only to a deeper understanding of the fundamental diffusion of these solutions, but also to a better understanding of the factors governing the formation of supramolecular structures.

In addition, the conductimetric technique has also been also used to measure mutual diffusion coefficients of solutions of ionic carriers and ionic drugs. A key advantage of conductimetric measurements over other methods is their ability to make accurate diffusion measurements for very dilute solutions relevant to therapeutic drug dosages. The results provide rates of diffusion of the free drug and free carrier molecules for comparison with the rates of facilitated diffusion of the drug–carrier complexes. Moreover, diffusion data coefficients for dilute solutions have be analyzed to determine thermodynamic activity coefficient data not obtainable by other techniques. Furthermore, a detailed understanding of the results, including the mechanism of coupled drug–carrier diffusion, has been provided with the help from Leaist et al. [30,31,32].

Density and viscosity values of the solutions containing drugs and carriers, complemented with the diffusion experiments, provide partial molar volume values of the solution components. This information is required to understand the thermodynamic behaviour and to estimate the sizes and friction coefficients of the various species diffusing in the solutions. Accurate viscosity measurement of drug–carrier solutions gives the necessary information for the development of the semi-empirical Stokes–Einstein models in these systems. 

## 2. Techniques: A Brief Description and the Analysis of Their Accuracy

### 2.1. Mutual Diffusion

Mutual differential isothermal diffusion coefficients of the electrolytes and non-electrolytes in aqueous solutions are of great interest not only for fundamental purposes, in helping to understand their nature, but also for many technical fields such as biomedical and pharmaceutical applications [25]. 

This research is justified by the lack of available diffusion data for pharmaceutical systems, especially multicomponent mutual diffusion coefficients describing the coupled transport of drugs and carriers, and by the difficulty in predicting theoretically accurate diffusion coefficients. In fact, the lack of diffusion coefficients in the scientific literature due to the difficulty of their accurate experimental measurement and impracticability of their accurate determination by theoretical procedures, allied to their industrial need, well justifies efforts in obtaining such accurate measurements.

At constant temperature, the concentration gradient inside a solution (without convection or migration) produces a flow of matter in the opposite direction to this gradient, which arises from random fluctuations in the positions of molecules in space. This phenomenon, known as isothermal diffusion, is an irreversible process. The gradient of chemical potential in real solutions is treated as the true virtual force producing diffusion. However, in ideal solutions, that force can be quantified by the gradient of the concentration at constant temperature. Thus, we consider the thermodynamics of irreversible processes and Fick’s laws to describe the isothermal diffusion [25]. The diffusion coefficient, *D*, in a binary system can be defined in terms of the concentration gradient by a phenomenological relationship known as Fick’s first law: (1)$$J=-D\frac{\partial c}{\partial x},$$
where *J* represents the flow of matter across a suitable chosen reference plane per area unit and per time unit, in a one-dimensional system, and *c* is the concentration of solute in moles per volume unit at the point considered. Equation (1) can be used to measure *D*. The diffusion coefficient can also be measured considering Fick’s second law:(2)$$\frac{\partial c}{\partial t}=\frac{\partial }{\partial x}\left(D\frac{\partial c}{\partial x}\right).$$

In general, the available methods are categorised into two groups: steady- and unsteady-state methods, according to Equations (1) and (2), respectively. In most processes, diffusion is a three-dimensional phenomenon. However, many of the experimental methods used to analyze diffusion restrict it to a one-dimensional process. Additionally, it is much easier to study their mathematical treatments in one dimension (which then can be generalized to a three-dimensional space).

The resolution of Equation (2) for a unidimensional process is much easier if we consider *D* as a constant. This approximation is applicable only when there are small differences in concentration, which is the case in our open-ended conductimetric technique (Figure 1) [1] and in the Taylor technique (Figure 2) [25]. In these circumstances, we may consider that all these measurements are parameters with a well-defined thermodynamic meaning. 

The conductimetric open-ended capillary cell technique (Figure 1), useful to measure mutual diffusion coefficients of electrolytic solutions, was also used because it can be applied with good precision (accuracy: 0.5%) in the case of very dilute solutions, as is the case of drugs at therapeutic dosages. Diffusion coefficient data for dilute solutions obtained in this way can be analysed to determine thermodynamic activity coefficient values that cannot be obtained by other techniques. In this technique, two vertical capillaries, each closed at one end by a platinum electrode, are positioned one above the other with the open ends facing each other. The upper and lower tubes, initially filled with solutions at concentrations 0.75 *c* and 1.25 *c*, respectively, are surrounded with the bulk solution at *c* concentration. A central electrode (a platinum wire) is placed in between both electrodes covered by the bulk solution. In this way, the physical length of the capillary tube coincides with the diffusion path. The differential diffusion coefficient is obtained by measuring the rate of change of the resistances between the top (or the bottom) electrode and the central one (at ground potential). 

### 2.2. Conductivity Measurements 

Conductivity measurements for solutions of ionic drugs and carriers (e.g., ethambutol dichloride and the ionic surfactant sodium dodecylsulfate) have been measured with a Wayne Kerr automatic bridge (frequency: 1 kHz). The electrical resistance of the conductimetric cells (Shedlovsky type) has been converted to specific conductivities by using homemade software. The solute-adding method has been used to change the solution’s concentration in the cell. From these conductivity measurements, ionic mobilities and diffusion coefficient values have been obtained for comparison with the experimental values. From these measurements, it is possible to calculate the limiting electrolyte diffusion coefficients in cases where the limiting counterion conductivity is known. 

Molar conductivities extrapolated to zero concentration provided accurate limiting diffusion coefficients (Nernst coefficients) of the drugs, carriers, and counterions. This information is essential for Nernst–Planck models of the diffusion of ionic drugs and carriers, which include migrational fluxes driven by the electric field (diffusion potential) generated by the diffusion of ionic species. 

### 2.3. Densities and Viscosities

We have obtained accurate (i.e., better than 0.01%) densities of solutions of drugs and carriers. Analysis of this data yields partial molar volumes of the solution components. This information is required to understand the thermodynamic behaviour and to estimate the sizes and friction coefficients of the various species diffusing in the solutions. The diffusion measurements have been complemented by these data. Densities of the solutions used in our diffusion experiments have been measured with a densimeter of vibrating U-tube Anton Paar DMA5000 (reproducibility within ±5 × 10^−6^ g·cm^−3^ and uncertainty of ±5 × 10^−5^ g·cm^−3^). Analysis of the density data provided partial molar volumes of the drugs, carrier molecules, and the solvent. A more detailed description will be found in the following chapters.

Viscosity measurements have been performed with an Ostwald-type viscometer, kept in a transparent-walled circulating water bath supplied with a heating/cooling system to maintain thermal equilibrium. The viscosity coefficient, *η*, is determined from the elapsed time that the solution flows through two calibrated marks on the capillary of the viscometer. Viscosity data are essential in order to understand rates of transport in the solutions. 

## 3. Biological Systems of Interest 

The development and characterization of controlled-release systems for the reliable and safe delivery of precise dosages of drugs is an active area of our research (e.g., [13,15,21]). Cyclodextrins (Figure 3) and their derivatives [36] and block copolymer micelles [37] are among the most important carriers used in drug release systems for local anaesthetics and other pharmaceuticals. The basis for this popularity is the ability of these materials to solubilize poorly soluble drugs, resulting in striking increases in their water solubilities and rates of diffusion. 

Resorcinarenes (RAs) and resorcinarene derivatives are a class of macrocycles that form complexes with a variety of compounds through supramolecular interactions (Figure 4) [38,39,40,41]. In biomedical and pharmaceutical applications, RAs are used to host drugs, providing sustained-release dosages of these drugs for amyloid-β fibrillation inhibition [42] and other applications [43].

## 4. Deliverables

Our studies on the isothermal diffusion of drugs have permitted us to obtain a better understanding of the structure and transport behaviour of these systems, supplying the scientific and technological communities with data about these important parameters in solution. Among them, we have obtained new thermodynamic data (e.g., association constants, binding constants, dissociation degrees, activity coefficients, and thermodynamic factors), transport data (e.g., diffusion coefficients at infinitesimal concentrations, activation energies, ionic conductivity, and mobility factors), and structural parameters (e.g., effective hydrodynamic radii). The results have provided new information about molecular motions and interactions, helping us to improve information about the rates of chemical and physical processes of practical significance, such as diffusion-limited reactions, carrier-mediated transport, solubilization, gas absorption, crystal growth, chemical waves and oscillations, and diffusion driven by temperature gradients (i.e., Soret effect). This work delivers new fundamental information about the properties of micelle carrier systems, with different applications in different areas (e.g., industrial and biological). In addition, we studied the analysis of the results, the development of theoretical models to explain coupled diffusion behaviour, and the molecular structures in associating systems.

In summary, this work illustrates the significant research contributions concerning the thermodynamic and transport properties of systems involving drugs and carriers.

## Figures and Tables

**Figure 1 biomolecules-08-00178-f001:**
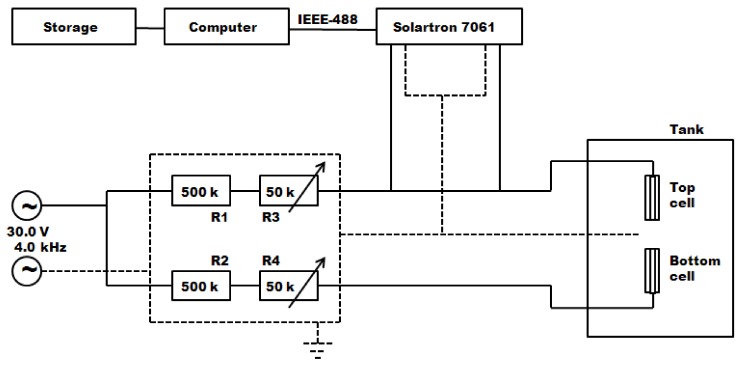
Scheme of the open-ended conductimetric technique [1]. R1, R2, R3 and R4 are resistances.

**Figure 2 biomolecules-08-00178-f002:**
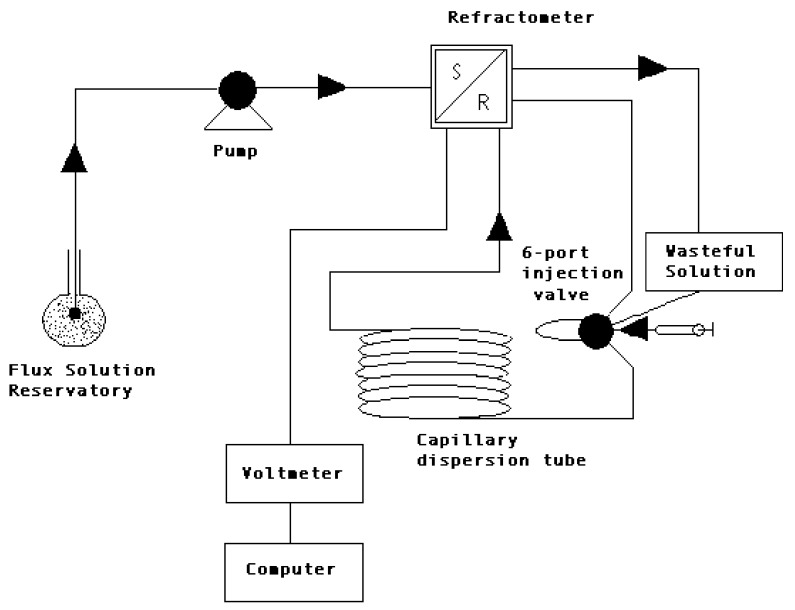
Schematic representation of the Taylor dispersion technique [25]. One of the main techniques used in the characterization of diffusion is the Taylor dispersion technique (accuracy: 1%–2%), a rapid and convenient flow technique to measure mutual diffusion in solutions. Multicomponent mutual diffusion (chemical interdiffusion) coefficients for both electrolyte and nonelectrolyte drugs and carriers have been determined by using the Taylor dispersion technique. This technique is based on the dispersion of small amounts of solutes injected into carrier solutions flowing through a capillary tube. The combined action of radial diffusion and convection along the tube axis causes the injected solute samples to spread out, producing Gaussian concentration profiles. Mutual diffusion coefficients, including cross-coefficients describing coupled transport, are calculated from refractive index profiles measured across the dispersed solute peaks at the outlet of the dispersion tube, employing conventional pulse-injection Taylor experiments and step-function techniques developed for studies of dilute solutions. R: reference cell, S: sample cell.

**Figure 3 biomolecules-08-00178-f003:**
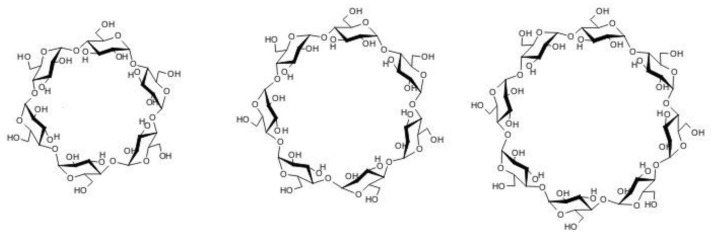
α-, β-, and γ-cyclodextrins’ structure.

**Figure 4 biomolecules-08-00178-f004:**
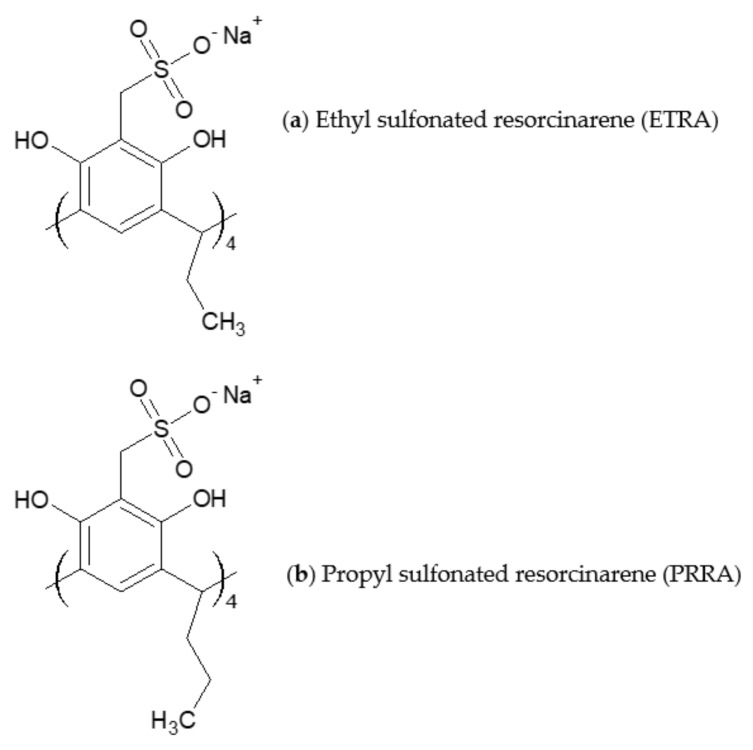
Molecular structures of the corresponding monomers of (**a**) tetrasodium 5,11,17,23-tetrakissulfonatemethylen-2,8,14,20-tetra(ethyl)resorcin[4]arene (ETRA) and (**b**) tetrasodium 5,11,17,23-tetrakissulfonatemethylen-2,8,14,20-tetra(propyl)resorcin[4]arene (PRRA).

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
