# Peer review of "Transport Properties for Pharmaceutical Controlled-Release Systems: A Brief Review of the Importance of Their Study in Biological Systems"

_biomolecules, 2018, doi:10.3390/biom8040178_

Reviewer 1 Report

The reviewed article is interesting from biomolecules point of view and theme of the article meets the scope of the journal. Work is performed at sufficient scientific level and has good quality; the results of investigation are professionally interpreted.  I suggest its publication by biomolecules after minor revision. The following are some minor comments for the authors to improve the quality of their manuscript.

Line 34: "give" to "given"

Line 53: "by" to "for"

Line 54: There is no need for the "" in "in vivo".

Line 55: remove "as".

Line 59: "the understanding of" to "to understand".

Line 60: "releasing" for "release".

Line 60: add a comma after "both".

Line 69: "for systems" should read "of systems".

Line 69: "its" to "their".

Line 74: "e.g." should be in italics.

Line 78: The "'s" in Ribeiro's is unnecessary.

Line 81: "study" to "studied".

Line 83: add comma after "both"

Line 84: add comma after "interactions"

Line 107: "and not yet studied" to "have not yet been studied".

Line 108: add comma after "groups"

Line 108, remove extra space after "it"

Line 115: "system" to "systems"

Line 115: the phrase "applications to mixing processes" is very confusing. I do not understand what you mean. Please rephrase.

Line 127: the phrase "(many times hysiological barriers)" is confusing and not clear. Please rephrase.

Line 128: "have" to has".

Line 129: "its" to "their".

Line 130: add comma after "both".

Line 131: add comma after "properties"

Line 149: please choose another word to replace "Moreover" as it is repeated from the previous line.

Line 151: "et al." should be in italics.

Line 156: please add comma after "measured"

Line 158: "of them" is unnecessary

Line 161: "its" for "their"

Line 171: "denominated by" is incorrect. Please rephrase.

Line 226: "have" to "has".

Line 226: "solution" to "solution's".

Line 231: "provide" to "provided".

Line 243: "provide" to "provided".

Line 256: "poorly soluble" to "poorly-soluble".

Line 293: "with applications to mixing processes" is unclear. Please rephrase. 

The references 3 and 4 should be in the same format

Author Response

We are grateful for the reviewer for his/her comments. The referee is right and, consequently, we have checked the whole text and changes have been made. The new version of the ms, in which the modifications introduced are highlighted in blue for the Reviewer 1.

Reviewer 2 Report

The authors attempts to provide a comprehensive review on the diffusion properties of drugs in the presence of host carriers. I believe that remarking the importance of cross-diffusion effects is very important, especially for these complex drug formulations. Thus, I support the main goal of this manuscript. However, the authors need to stress that they are specifically discussing their work, and also mention other related works.

For example, their statement

“Surprisingly, except for two studies of host-guest diffusion in aqueous cyclodextrin solutions, multicomponent mutual diffusion data for controlled-release systems are non-existent”

is remarkably inaccurate. A cursory literature search on mutual diffusion and cyclodextrin show several papers on cyclodextrin cross-diffusion effects including

Paduano J Phys Chem 1990 6885

Paduano J Sol Chem 1995 1143

Moreover, on controlled release and the role of cross-diffusion effects, there is the following paper in terms of an equivalent frictional formalism:

Wesselingh Controlling Diffusion JCR 1993 47

More recently there are two papers investigating cross-diffusion effects for drug molecules in the presence of carriers with related theories:

Zhang Langmuir 2008 10680

Zhang Langmuir 2009 3425

I think that it may be acceptable if the authors want to focus on their own work, but they should recognize and somewhat discuss the existence of these other experimental/theoretical studies and related experimental techniques. Especially if this work is labeled as a review.

Minor

In Abstract Line 18: Drug carriers are expected to reduce not increase drug diffusion due to their larger size. Their statement needs to be clarified.

Author Response

1.We are grateful for the reviewer for his/her comments. The reviewer is right when he says that we should recognize and somewhat discuss the existence of these other experimental/theoretical studies and related experimental techniques. In fact, the way the authors intend to sensitize the problem of lack of transport data of these systems, contrary to other proprieties (e.g. thermodynamic), was not the happiest. They would like to inform to the scientific community that while numerous studies have been carried out on these systems (e.g thermodynamic),  few have taken into account the transport behaviour of the complex chemical systems.

Consequently, we have modified the text and inserted some references, highlighting some works in this area.That is, we have modified the following sentence,

“Surprisingly, except for two studies of host-guest diffusion in aqueous cyclodextrin solutions, multicomponent mutual diffusion data for controlled-release systems are non-existent”

by

“While numerous studies have been carried out on these systems, few have taken into account the transport behavior for controlled-release systems.”

2. The reviewer is right and, consequently, we have modified the text, accordingly.

That is,

“The basis for this popularity is the ability of these materials to solubilize poorly soluble drugs and drug candidates, resulting in striking increases in their water solubilities and rates of diffusion”

 by

“The basis for this popularity is the ability of these materials to solubilize poorly soluble drugs, result in striking increases, in general, in their water solubilities”

In fact, these carriers (e.g., Ăź-CD) have the capacity to increase the solubility of some drugs. For example, there are L-dopa-based pharmaceutical formulations taking advantage of the CD complex properties have been developed for oral administration. However, in the case of HP-Ăź-CD (other carrier), there was no significant increase in the L-dopa solubility, suggesting a low binding constant for L-dopa-CD, which was later confirmed by Barros et al. using ternary interdiffusion coefficients (K=24M), whereas with 2,6-dimethyl-Ăź-cyclodextrin a significant increase in the L-dopa-CD host-guest association has been found. The formation of L-dopa supramolecular structures with Ăź-CDs, involving a higher association constant (ca. 2000 M) has been found to limit hydrolysis and photo degradation, thus, increasing the shelf life and the bioavailability, while also improving drug delivery.

The new version of the ms, in which the modifications introduced are highlighted in yellow for the Reviewer 2.

Round  2

Reviewer 1 Report

The authors taken in account all the suggestions

Author Response

We are grateful for the reviewer for his/her comments.

Reviewer 2 Report

The authors made an effort to include more references on drug diffusion and cross-diffusion effects. My remaining concern is that their current introductory statements in their "literature review" section, referring to transport and thermodynamic properties, is too vague and generic. The authors should specify that work from these other references also focused on multicomponent diffusion including coupled transport. Although it is expected that authors focus on the Taylor's method, other methods such as interferometric methods, should be mentioned in a section denoted as "literature review". The authors can specifically claim in the manuscript that they will focus on the Taylor's method, recognizing that interferometric methods have also been used to determine multicomponent diffusion coefficients for ternary systems relevant to diffusion-based controlled release.  There is also some typos related to references. There is a jump from ref 8 to ref 11 (refs 9,10 are not cited). At the very end of their file, refs 8,9 should be separated while ref 9 should become ref 10 and so on.

Author Response

1a) We are grateful for the reviewer for his/her comments. The reviewer is completly right when he says that we should recognize that in our current introductory statements in our "literature review" section, referring to transport and thermodynamic properties, is too vague and generic, and, also, we agree with him when he said that although it is expected that authors focus on the Taylor's method, other methods such as interferometric methods, should be mentioned in a section denoted as "literature review".

Consequently, we have modified the text. That is, we have modified the following sentence,

“Our research group and others, in the last two decades [1-8], has studied the transport and thermodynamic properties of systems involving drugs motivated by their practical contribution to a better understanding of the mechanism of drug release….”

by

“Our research group and others, in the last two decades [1-9], has studied the transport properties of systems involving drugs motivated by their practical contribution to a better understanding of the mechanism of drug release. The work developed by these authors [1-9], has been focused on multicomponent diffusion including coupled transport, by using different methods (that is, Taylor's method with acceptable uncertainty of 1–2%, and the interferometric methods with uncertainty<0.5%). In fact, both methods have also been used to determine multicomponent diffusion coefficients for ternary systems relevant to diffusion-based controlled release. However, in the present work, it will focus on the Taylor's method….”

1b) The reviewer is right and, consequently, we have modified the text, accordingly.